# Impaired Modulation of the Autonomic Nervous System in Adult Patients with Major Depressive Disorder

**DOI:** 10.3390/biomedicines12061268

**Published:** 2024-06-06

**Authors:** Elise Böttcher, Lisa Sofie Schreiber, David Wozniak, Erik Scheller, Frank M. Schmidt, Johann Otto Pelz

**Affiliations:** 1Department of Psychiatry and Psychotherapy, University Hospital Leipzig, Semmelweisstraße 10, 04103 Leipzig, Germany; 2Department of Neurology, University Hospital Leipzig, Liebigstraße 20, 04103 Leipzig, Germany

**Keywords:** major depressive disorder, heart rate variability, HRV, autonomic nervous system, parasympathetic nervous system, vagal tone, depression, sympathetic hyperactivity, modulation

## Abstract

Patients with major depressive disorder (MDD) have an increased risk for cardiac events. This is partly attributed to a disbalance of the autonomic nervous system (ANS) indicated by a reduced vagal tone and a (relative) sympathetic hyperactivity. However, in most studies, heart rate variability (HRV) was only examined while resting. So far, it remains unclear whether the dysbalance of the ANS in patients with MDD is restricted to resting or whether it is also evident during sympathetic and parasympathetic activation. The aim of this study was to compare the responses of the ANS to challenges that stimulated the sympathetic and, respectively, the parasympathetic nervous systems in patients with MDD. Forty-six patients with MDD (female 27 (58.7%), mean age 44 ± 17 years) and 46 healthy controls (female 26 (56.5%), mean age 44 ± 20 years) underwent measurement of time- and frequency-dependent domains of HRV at rest, while standing (sympathetic challenge), and during slow-paced breathing (SPB, vagal, i.e., parasympathetic challenge). Patients with MDD showed a higher heart rate, a reduced HRV, and a diminished vagal tone during resting, standing, and SPB compared to controls. Patients with MDD and controls responded similarly to sympathetic and vagal activation. However, the extent of modulation of the ANS was impaired in patients with MDD, who showed a reduced decrease in the vagal tone but also a reduced increase in sympathetic activity when switching from resting to standing. Assessing changes in the ANS during sympathetic and vagal activation via respective challenges might serve as a future biomarker and help to allocate patients with MDD to therapies like HRV biofeedback and psychotherapy that were recently found to modulate the vagal tone.

## 1. Introduction

Depression is a major burden of disease with a high risk for recurrence and chronification [1]. The worldwide prevalence of major depressive disorders (MDDs) of 4.2% in women and 2.7% in men is even increasing [1]. MDD is not only a mood disorder, but body health, especially metabolic, hepatic, and immune health, is often compromised in patients with MDD [2]. Moreover, patients with MDD also have an increased risk for cardiac events, whose cause is hitherto unclear [3]. Over the last decades, a dysregulation of the autonomic nervous system (ANS) has aroused increasing attention in MDD. Generally, despite an existing heterogeneity across included populations, underlying diseases, study features, and heart rate variability (HRV) parameters, lower measures of HRV at study inclusion were associated with an increased mortality from any cardiac cause [4]. Measures of HRV robustly demonstrated a reduced vagal, i.e., parasympathetic, tone, resulting in a relative sympathetic hyperactivity in patients with MDD [5]. In their meta-analysis, Koch and colleagues compared resting-state measures of HRV between 2250 unmedicated adults with MDD and 1982 controls without coronary heart disease (CHD) or cardiac medication and found that all (frequency- and time-dependent) HRV measures were lower in patients with MDD than in healthy controls [6]. Brown and colleagues replicated these findings of a reduced HRV in a meta-analysis in older adults (over 60 years), including 550 patients with MDD without CHD compared to healthy controls [7]. This sympatho-vagal dysbalance is also assumed to contribute to the increased risk for CHD and even mortality in patients with MDD [8,9,10].

However, in most HRV studies, patients with MDD and controls were only examined while resting [6]. Thus, it remains unclear whether the reduced vagal tone is restricted to the resting state or whether it is also evident during sympathetic or parasympathetic activation of the ANS and whether relative changes between resting and activation are different between patients with MDD and controls.

The aim of this study was to compare the responses of the ANS in patients with MDD to challenges that stimulated the sympathetic and, respectively, the parasympathetic nervous systems. Due to the bidirectional relationship between MDD and CHD, given that MDD is a relevant risk factor for CHD and that patients with CHD and MDD often showed a more pronounced affection for the ANS than CHD patients without MDD [9,11], we focused on patients with MDD without known CHD.

## 2. Methods and Materials

### 2.1. Study Design

This study was performed according to the ethical standards laid down in the 1964 Declaration of Helsinki and its later amendments. It was approved by the local ethics committee of the medical faculty at the University of Leipzig (reference number 425/19-ek). All participants gave informed and written consent for their participation.

Participants were recruited from June 2020 to September 2021 from the Department of Psychiatry and Psychotherapy, University Hospital Leipzig. All patients had to fulfill the clinical criteria of depression in accordance with the International Statistical Classification of Diseases and Related Health Problems, 10th Revision (ICD 10: F32.1–F32.2 and F33.1–F33.3).

Exclusion criteria were the use of illegal substances, any psychiatric diagnoses in the control group, organic or psychotic psychiatric comorbidities in the MDD group, and a history of head injury, polyneuropathy, epilepsy, neurodegenerative disorders, relevant cardiovascular comorbidities like atrial fibrillation or known coronary heart disease, or acute somatic diagnoses at the time of examination in either group. In addition, all participants underwent a profound neurological examination to exclude those with clinically apparent yet hitherto unknown polyneuropathy. To evaluate the severity of depression at the time of participation, all participants completed the revised Beck Depression Inventory (BDI–II).

### 2.2. Measurement of Heart Rate Variability

After at least 5 min of resting in a calm environment, i.e., in a dimly lit, quiet room with controlled temperature, RR intervals were measured under normal breathing for 5 min while resting and in a supine position with the upper part of the body elevated to 30°. Subsequently, participants stood up, and measurement of HRV was repeated for 5 min of standing. Finally, for measurements of HRV under slow-paced breathing (SPB), participants sat on a chair and breathed with a frequency of 6 breaths per minute (6 s of inhalation followed by 4 s of exhalation). Slow-paced breathing was visualized by a metronome (rising and falling bar) on a computer monitor. Altogether, 120 RR intervals were recorded during SPB. During all measurements, participants were advised to relax, not to move, and not to speak.

For recording the electrocardiogram (ECG), electrodes were fastened at both wrists and ankles. Analyses of HRV were carried out by the computer-based system ProSciCard (MedSet Medizintechnik GmbH, Hamburg, Germany). Besides heart rate, several indices of the HRV were automatically computed: the standard deviation of RR intervals (SDNN, higher values indicate higher variability) and the root mean square of successive differences (RMSSD, estimate of short-term components of the HRV). In addition, power spectral analyses in the low- (ln(LF), 0.05–0.15 Hz) and high-frequency spectrums (ln(HF), 0.15–0.5 Hz) were performed, and the low-frequency/high-frequency ratio (ln(LF/HF) ratio) was calculated. Power spectral analyses were calculated while lying and standing, but due to restrictions of the proprietary software, they were not performed during SPB.

After each measurement, the recorded ECG was checked for artifacts by the examiner, and the identified artifacts were excluded from further analyses.

### 2.3. Statistical Analysis

Statistical analyses were performed by using IBM SPSS Statistics (IBM Corporation, Armonk, New York, USA; version 29.0). Explorative analysis of the HRV parameters revealed extreme outliers that lacked biological plausibility. Thus, extreme outliers were identified and excluded before performing further statistics based on Tukey’s hinges (first quartile − 3 × interquartile range (IQR) and third quartile + 3 × IQR) and visualized in boxplots [12]. Mann–Whitney U test was used for intergroup comparison of independent variables, while Wilcoxon signed-rank test was used for comparison of dependent variables (intragroup comparison). Chi-square test and Fisher’s exact test were applied for group comparisons of nominally scaled data. The significance level was set at *p* < 0.05.

## 3. Results

Demographic data of patients with MDD and the control group were well-balanced in terms of sex, age, height, weight, and BMI. Smoking was more frequent in patients with MDD. The prevalence of further cardiovascular risk factors and relevant comorbidities was similar between both groups (Table 1).

At rest, patients with MDD showed a higher heart rate (70.4 ± 12.2 vs. 64.1 ± 9.3 beats per minute, Mann–Whitney U, *p* = 0.013), a reduced HRV (SDNN, 37.6 ± 16.2 vs. 54.7 ± 27.8 ms, Mann–Whitney U, *p* = 0.003), and a reduced vagal tone (RMSSD, 30.9 ± 15.7 vs. 46.3 ± 27.3 ms, Mann–Whitney U, *p* = 0.003) compared to controls. The power spectral analyses in the low-frequency spectrum (ln(LF)), indicating the sympathetic activity, were even higher in controls than in MDD patients (4.9 ± 6.2 vs. 4.5 ± 11.7, Mann–Whitney U, *p* = 0.037; Table 2).

While standing, heart rate tended to be higher in patients with MDD (85.4 ± 17.2 vs. 80.0 ± 12.7 beats per minute, Mann–Whitney U, *p* = 0.086). HRV (SDNN, 32.4 ± 14.0 vs. 45.5 ± 21.1 ms, Mann–Whitney U, *p* = 0.002) was lower in patients with MDD, while no changes were observed for the vagal tone (RMSSD, 22.1 ± 9.7 vs. 27.2 ± 14.5 ms, Mann–Whitney U, *p* = 0.157). Again, the sympathetic activity (ln(LF) was higher in controls (12.4 ± 15.6 vs. 5.6 ± 8.5, Mann–Whitney U, *p* = 0.012). During SPB, heart rate tended to be higher in MDD patients (74.3 ± 11.6 vs. 69.4 ± 9.3 beats per minute, Mann–Whitney U, *p* = 0.060), while HRV (SDNN, 62.1 ± 32.6 vs. 85.5 ± 44.6 ms, Mann–Whitney U, *p* = 0.014) and vagal tone (RMSSD, 38.8 ± 18.7 vs. 51.7 ± 26.3 ms, Mann–Whitney U, *p* = 0.038) were lower (Table 2).

Analyzing intragroup changes while switching from resting to standing, patients with MDD and controls showed a similar pattern of sympathetic activation with an increase in heart rate, a decrease in HRV (SDNN), a decrease in vagal tone (RMSSD), and an increase in ln(LF) (Table 3 and Table 4). Comparing resting state with SPB, there were again similar patterns in both groups of parasympathetic activation with a higher vagal tone (RMSSD), a higher HRV (SDNN), as well as a higher heart rate (Table 3 and Table 4).

When switching from resting to standing, the decrease in the vagal tone (RMSSD) was significantly lower in patients with MDD than in healthy controls (ΔRMSSD, −8.8 ± 16.9 vs. −19.0 ± 23.2 ms, Mann–Whitney U, *p* = 0.014), while the increases in the sympathetic activity (Δln(LF), 7.5 ± 14.4 vs. 1.2 ± 12.1, Mann–Whitney U, *p* = 0.026) and the sympatho-vagal balance (Δln(LF/HF), 1.8 ± 5.2 vs. 1.1 ± 2.2, Mann–Whitney U, *p* = 0.002) were higher in controls (Table 5).

The MDD group was further stratified into two subgroups: first depressive disorder (FDD, 17 patients) and recurrent depressive disorder (RDD, 29 patients). There were no differences in any parameter of HRV between FDD and RDD patients. We also found no correlation between the BDI-II sum score, the duration of the current depressive episode, or the time in hospital and HRV parameters.

## 4. Discussion

Consistent with recent studies [13,14], we found that despite the relative sympathetic hyperactivity at rest in MDD patients, principal mechanisms of activation of the sympathetic nervous system while standing and activation of the parasympathetic nervous system during SPB were similar between patients with MDD and controls. However, the extent of this sympathetic and parasympathetic modulation was lower in MDD patients than in controls.

A relative sympathetic hyperactivity at rest in patients with MDD was indicated by a higher heart rate and lower HRV (SDNN), as well as a reduced vagal tone (lower RMSSD). While standing, the relative sympathetic hyperactivity decreased but was still evident with a reduced HRV (SDNN) and a trend of a higher heart rate in MDD patients compared to controls. Interestingly, when switching from resting to standing, the decrease in the vagal tone (RMSSD) was significantly lower in patients with MDD, while the increase in the sympathetic activity (ln(LF)) was higher in controls. The vagal tone (RMSSD) during SPB was still significantly higher in controls compared to MDD patients. Both, patients with MDD and controls responded to the parasympathetic challenge with a significant increase in HRV (SDNN), while the increase in the vagal tone (RMSSD) was only significant in depressive patients. The extent of the modulation of the ANS upon a parasympathetic stimulus (SPB) was similar in patients with MDD and controls.

So far, only a few studies have examined HRV in MDD not only at rest but also during sympathetic and parasympathetic challenges. Like in this study, Pradeep and colleagues reported an impaired parasympathetic modulation when comparing frequency-dependent HRV parameters (HF, LF, LF/HF ratio) that were measured during lying and in response to a physiological maneuver (standing) in 46 drug-naïve patients with MDD (mean age 36 years) and respective controls [15]. Noteworthy, the drop in HF from the supine position to standing, as an indicator of parasympathetic activity, was also significantly higher in the control group as compared to the MDD group [15]. These findings might suggest that in patients with MDD, the vagal tone is already in the lower range even while resting. Similar to the findings in our study, a lower vagal tone at rest, while standing, and during SPB was reported in two recent studies with 40 depressive patients (mean age 35 years, 62.5% females; [13]) and, respectively, 91 drug-naïve patients with MDD (mean age 30 years, 36.3% females; [14]) that both measured frequency- and time-dependent HRV parameters. However, neither study analyzed the relative changes in HRV parameters in MDD patients and controls between resting and the respective autonomic challenges. Thus, no conclusions can be drawn from these studies regarding the extent and possible differences in the autonomic modulation between groups [13,14].

In general, the treatment of MDD involves a multi-modal approach with antidepressant medication, psychological therapy, psychoeducation, sports, and electroconvulsive therapy [16]. The modulation of the ANS in MDD patients with the aim of increasing the vagal tone might improve depressive symptoms but also exhibit beneficial effects on the cardiovascular system, e.g., [17,18,19]. There are several non-pharmacological approaches to improve HRV in patients. One way to increase the vagal tone is using HRV biofeedback with visualization of SPB [20]. A pilot study found an improvement in depressive symptoms, a reduced heart rate, and increased HRV in patients with MDD after HRV biofeedback but not in the control group [18]. Cognitive behavior therapy also resulted in a decrease in heart rate and an increase in the vagal tone (indicated by an increase in RMSSD) in both patients with severe depression and stable CHD [17] and in MDD patients without CHD [21]. The combination of HRV biofeedback and psychotherapy over a six-week period even showed a larger increase in HRV and a larger decrease in depressive symptoms relative to a group that was only treated with psychotherapy [22]. Finally, regular aerobic exercise over 16 weeks improved depressive symptoms and HRV in patients with CHD and the subset of patients who were diagnosed with MDD at study entry [19]. It may take weeks until the effect of non-pharmacological therapies and antidepressant medication can be observed [16]. Thus, initially, the modulation of the ANS in MDD by respective challenges might serve as an easily assessable biomarker that could help to predict treatment response and, if confirmed, to allocate patients to specific non-pharmacological therapies. However, so far, all the above-mentioned studies were single-center studies and differed with respect to the number of included MDD patients, age and gender of patients, and concurrent antidepressive medication. Hence, replication in larger, multi-center trials is needed.

A reduced HRV is not exclusive to MDD but is also found in many psychiatric and neurological diseases like anxiety disorder, amyotrophic lateral sclerosis, or Parkinson’s disease [20,23]. Examining the morphology of the vagus nerve, an inverse correlation between the cross-sectional area (CSA) of the left vagus nerve and SDNN (HRV) and RMSSD (vagal tone) was found in healthy subjects. That means the larger the left vagus nerve CSA, the lower the HRV and the lower the vagal tone [24]. Notably, an increased cross-sectional area of the left vagus nerve was found in patients with MDD [25], and this enlargement of the left vagus nerve was hypothesized to be due to a chronic low-grade inflammation [25]. Chronic stress is a risk factor for MDD and, in particular, for the melancholic phenotype of depression [26]. Chronic stress is linked to a dysregulation of the hypothalamic–pituitary–adrenal (HPA) axis with, amongst others, an increased release of cortisol, resulting in the hyperactivation of the peripheral immune system and low-grade inflammation [26]. Via the (left) vagus nerve, this systemic low-grade inflammation might contribute to the dysregulation of the ANS. A hyperfunction of the HPA axis can also alter cardiac autonomic function with sympathetic overactivation and parasympathetic hypofunction. This imbalance of the cardiac autonomic function may result in an increased heart rate, an increased myocardial contractility, and an increased myocardial oxygen consumption, which finally makes the heart more susceptible to malignant cardiac arrhythmias [11,27].

### Limitations

This study has several limitations. First, the mean interval from admittance to hospital to study enrollment was 17 days. Since the BDI-II was assessed at study enrollment, it might have changed in the meantime because of the beginning of antidepressive therapies prior to study enrollment. This might explain why we did not find correlations between the BDI-II score or the time in hospital and any HRV parameter. Secondly, all patients were treated with antidepressants, psychological therapy, and sports therapy at the time of HRV assessment. Thus, we cannot rule out the effect of the antidepressive treatment on HRV parameters. However, since psychological therapy and sports were reported to increase the vagal tone [19,21], they just might mitigate but not reverse the effects that were found in this study. Finally, we examined patients with their first but also with recurrent depressive episodes. Both subgroups were too small for separate statistical analyses.

## 5. Conclusions

Patients with MDD and controls responded similarly to sympathetic and vagal activation. However, the extent of modulation of the ANS was impaired in patients with MDD, who showed a reduced decrease in the vagal tone but also a reduced increase in the sympathetic activity when switching from resting to standing. Assessing changes in the ANS during sympathetic and vagal activation via respective challenges might help to allocate patients with MDD to therapies like HRV biofeedback and psychotherapy that were recently found to modulate the vagal tone.

## Figures and Tables

**Table 1 biomedicines-12-01268-t001:** Demographic data of patients with major depressive disorder (MDD) and healthy controls. ^#^ Chi-square test; ° Fisher’s exact test; ^+^ Mann–Whitney U test. * One extreme outlier of 400 weeks duration was excluded. Significant *p* values are in bold. SD: standard deviation; BMI: body mass index; BDI-II: revised Beck Depression Inventory.

	MDD Group (*n* = 46)	Control Group (*n* = 46)	*p*-Value
Female (*n* (%))	27 (58.7%)	26 (56.5%)	0.833 ^#^
Age in years (mean ± SD)	44 ± 17	44 ± 20	0.656 ^+^
Height in cm (mean ± SD)	172 ± 10	173 ± 10	0.809 ^+^
Weight in kg (mean ± SD)	79 ± 19	73 ± 12	0.080 ^+^
BMI in kg/m^2^ (mean ± SD)	26.5 ± 5.8	24.4 ± 3.6	0.061 ^+^
Cardiovascular risk factors and medical history of comorbidities			
Diabetes mellitus (*n* (%))	2 (4.3%)	0	0.495 °
Arterial hypertension (*n* (%))	8 (17.4%)	6 (13.0%)	0.562 ^#^
Smoking (*n* (%))	18 (39.1%)	9 (19.6%)	**0.039** ^#^
Sleep apnea	2 (4.3%)	0	0.495 °
Asthma	4 (8.7%)	3 (6.5%)	1.0 °
Hypothyroidism	6 (13.0)	3 (6.5%)	0.485 °
Questionnaires			
BDI-II sum score (mean ± SD)	24.4 ± 10.3	4.9 ± 4.2	**<0.001** ^+^
Duration of actual depressive episode in weeks (mean ± SD)	26 * ± 19	-	-
Time in hospital in days (mean ± SD)	46 ± 34	-	-

**Table 2 biomedicines-12-01268-t002:** Comparison of heart rate variability parameters between patients with major depressive disorder (MDD) and healthy controls while resting, standing (sympathetic challenge), and slow-paced breathing (SPB; parasympathetic challenge). Mann–Whitney U test was used for statistical comparison between groups. Significant *p* values are in bold. SD: standard deviation; SDNN: standard deviation of RR intervals; RMSSD: root mean square of successive differences; ln(LF): power spectral analyses in the low-frequency spectrum; ln(HF): power spectral analyses in the high-frequency spectrum.

	Resting MDD Group	Resting Control Group	*p*	Standing MDD Group	Standing Control Group	*p*	SPB MDD Group	SPB Control Group	*p*
Heart rate in beats per minute (mean ± SD)	70.4 ± 12.2	64.1 ± 9.3	**0.013**	85.4 ± 17.2	80.0 ± 12.7	0.086	74.3 ± 11.6	69.4 ± 9.3	0.060
SDNN (ms, mean ± SD)	37.6 ± 16.2	54.7 ± 27.8	**0.003**	32.4 ± 14.0	45.5 ± 21.1	**0.002**	62.1 ± 32.6	85.5 ± 44.6	**0.014**
RMSSD (ms, mean ± SD)	30.9 ± 15.7	46.3 ± 27.3	**0.003**	22.1 ± 9.7	27.2 ± 14.5	0.157	38.8 ± 18.7	51.7 ± 26.3	**0.038**
ln(LF) (mean ± SD)	4.5 ± 11.7	4.9 ± 6.2	**0.037**	5.6 ± 8.5	12.4 ± 15.6	**0.012**	-	-	
ln(HF) (mean ± SD)	2.3 ± 1.8	4.0 ± 4.2	0.100	2.2 ± 2.0	2.9 ± 2.8	0.258	-	-	
ln(LF/HF) (mean ± SD)	1.2 ± 1.3	1.7 ± 4.5	0.306	2.3 ± 2.2	3.6 ± 2.7	**0.007**	-	-	

**Table 3 biomedicines-12-01268-t003:** Comparison of heart rate variability parameters within patients with major depressive disorder (MDD) while resting, standing (sympathetic challenge), and slow-paced breathing (parasympathetic challenge). Wilcoxon signed-rank test was used for statistical comparison between groups. Significant *p* values are in bold. SD: standard deviation; SDNN: standard deviation of RR intervals; RMSSD: root mean square of successive differences; ln(LF): power spectral analyses in the low-frequency spectrum; ln(HF): power spectral analyses in the high-frequency spectrum.

	Resting	Standing	*p*	Resting	Slow-Paced Breathing	*p*
Heart rate in beats per minute (mean ± SD)	70.4 ± 12.2	85.4 ± 17.2	**<0.001**	70.4 ± 12.2	74.3 ± 11.6	**<0.001**
SDNN (ms, mean ± SD)	37.6 ± 16.2	32.4 ± 14.0	0.055	37.6 ± 16.2	62.1 ± 32.6	**<0.001**
RMSSD (ms, mean ± SD)	30.9 ± 15.7	22.1 ± 9.7	**0.003**	30.9 ± 15.7	38.8 ± 18.7	**0.004**
ln(LF) (mean ± SD)	4.5 ± 11.7	5.6 ± 8.5	**0.006**	-	-	-
ln(HF) (mean ± SD)	2.3 ± 1.8	2.2 ± 2.0	0.424	-	-	-
ln(LF/HF) (mean ± SD)	1.2 ± 1.3	2.3 ± 2.2	**0.002**	-	-	-

**Table 4 biomedicines-12-01268-t004:** Comparison of heart rate variability parameters within healthy controls while resting, standing (sympathetic challenge), and slow-paced breathing (parasympathetic challenge). Wilcoxon signed-rank test was used for statistical comparison between groups. Significant *p* values are in bold. SD: standard deviation; SDNN: standard deviation of RR intervals; RMSSD: root mean square of successive differences; ln(LF): power spectral analyses in the low-frequency spectrum; ln(HF): power spectral analyses in the high-frequency spectrum.

	Resting	Standing	*p*	Resting	Slow-Paced Breathing	*p*
Heart rate in beats per minute (mean ± SD)	64.1 ± 9.3	80.0 ± 12.7	**<0.001**	64.1 ± 9.3	69.4 ± 9.3	**<0.001**
SDNN (ms, mean ± SD)	54.7 ± 27.8	45.5 ± 21.1	**0.003**	54.7 ± 27.8	85.5 ± 44.6	**<0.001**
RMSSD (ms, mean ± SD)	46.3 ± 27.3	27.2 ± 14.5	**<0.001**	46.3 ± 27.3	51.7 ± 26.3	0.091
ln(LF) (mean ± SD)	4.9 ± 6.2	12.4 ± 15.6	**<0.001**	-	-	-
ln(HF) (mean ± SD)	4.0 ± 4.2	2.9 ± 2.8	0.067	-	-	-
ln(LF/HF) (mean ± SD)	1.7 ± 4.5	3.6 ± 2.7	**<0.001**	-	-	-

**Table 5 biomedicines-12-01268-t005:** Comparison of differences (Δ) of heart rate variability parameters between resting and a sympathetic challenge (standing), respectively, between resting and a parasympathetic challenge (slow-paced breathing) in patients with major depressive disorder (MDD) and healthy controls. Mann–Whitney U test was used for statistical comparison between groups. Significant *p* values are in bold. SD: standard deviation; SDNN: standard deviation of RR intervals; RMSSD: root mean square of successive differences; ln(LF): power spectral analyses in the low-frequency spectrum; ln(HF): power spectral analyses in the high-frequency spectrum.

	Δ Standing—Resting			Δ Slow-Paced Breathing—Resting		
	MDD Group	Control Group	*p*	MDD Group	Control Group	*p*
Heart rate in beats per minute (mean ± SD)	15.0 ± 8.9	13.9 ± 9.2	0.766	4.9 ± 5.2	5.1 ± 6.6	0.920
SDNN (ms, mean ± SD)	−5.2 ± 16.8	−9.2 ± 21.5	0.207	25.7 ± 29.8	31.2 ± 33.6	0.491
RMSSD (ms, mean ± SD)	−8.8 ± 16.9	−19.0 ± 23.2	**0.014**	9.2 ± 19.7	5.9 ± 24.1	0.558
ln(LF) (mean ± SD)	1.2 ± 12.1	7.5 ± 14.4	**0.026**	-	-	
ln(HF) (mean ± SD)	−0.1 ± 2.4	−1.1 ± 3.5	0.367	-	-	
ln(LF/HF) (mean ± SD)	1.1 ± 2.2	1.8 ± 5.2	**0.002**	-	-	

## Data Availability

Johann Otto Pelz, as the principal author, has full access to the data used in the analyses in the manuscript and takes full responsibility for the data, the analyses and interpretation, and the conduct of the research. The dataset underlying this study is available from the corresponding author upon reasonable request for any qualified investigator.

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
