# Peer review of "Impaired Modulation of the Autonomic Nervous System in Adult Patients with Major Depressive Disorder"

_biomedicines, 2024, doi:10.3390/biomedicines12061268_

Round 1
Reviewer 1 Report
Comments and Suggestions for Authors
STRUCTURE
- The title is related to the study
- The manuscript is not correctly structured.
TITLE AND ABSTRACT
- The Abstract is properly structured. The title is concise, specific and relevant. Type of patient is indicated.
- The title include the study population.
- Write better explaining the number of subjects and type of sample: women or men. In addition to better explaining what conclusions they reach in the study.
INTRODUCTION
- In general, the introduction is quite comprehensive, and shows a good overview of the current state of the subject.
- Very general introduction, that is, it explains that all the articles that have analyzed HRV and SDM have been at rest, but it does not explain any, what populations did they use? What type of intervention? How did they measure it? etc
MATERIAL AND METHODS
- This section includes different relevant data of the study without differentiating it by sections, it is advisable to separate it with the following sections: design, inclusion criteria, statistical analysis, etc.
2.2. Methods
- Line 78 - 101. Explain the reasons for inclusion and exclusion, in addition to the number of participants in the study, it would be advisable to make a diagram to make it clearer.
- Line 108. Used When the participants are standing actively, what do you mean? Are they walking or are they static?.
- Line 126 - 127. “Power spectral analyses were calculated while lying and standing, but not during SPB”. Why?
RESULTS
- The results are explained in order as they appear in the figures, well-detailed figures with footnotes and explained nomenclature.
- In Table 1, what the + symbol means is not explained in the nomenclature.
- The significances p<0.05 are not indicated with any symbol, so it is difficult to differentiate them from the significances p>0.05. It would be recommended that a symbol be added to the significances p<0.05 in the tables. and explain it in the table note.
- Table 1 is so complete that it is difficult to understand the results, separating it into 3 different tables. Because there is too much information and too many symbols.
- Table 1 it is difficult to differentiate the meanings.
- Table 2 it is difficult to differentiate the meanings.
- Table 3 it is difficult to differentiate the meanings.
- Table 4 it is difficult to differentiate the meanings.
- Table 5 it is difficult to differentiate the meanings.
DISCUSSION
- “Consistently with recent studies we found that despite the relative sympathetic hy-peractivity at rest in MDD patients, principal mechanisms of an activation of the sympa-thetic nervous system while standing and an activation of the parasympathetic nervous system during SPB were similar between patients with MDD and controls.” What studies? Give reference.
- “A relative sympathetic hyperactivity with decreased HRV was also repeatedly found in children, adolescents, and adults with MDD”. And a womans?
- “So far, only a few studies examined HRV in MDD not only at rest but also during sympathetic and parasympathetic challenges. Pradeep and colleagues found an attenu-ated response of frequency dependent HRV parameters to active standing in patients with MDD compared to healthy controls. They concluded that “there was an impaired para-sympathetic modulation in response to physiological maneuver (orthostatic challenge) in drug naïve subjects with major depression” (Pradeep et al., 2012). Similar to the findings in our study, a lower vagal tone at rest, while standing, and during SPB was reported in two recent studies (Chen et al., 2017; Kuang et al., 2019). However, both studies did not analyze relative changes of HRV parameters in MDD patients and controls between rest-ing and the respective autonomic challenges. Thus, no conclusions can be drawn from these studies regarding the extend and possible differences in the autonomic modulation between groups (Chen et al., 2017; Kuang et al., 2019)”. What type of intervention, how long and what subjects are used (age, sex).
- “The modulation of the ANS in MDD patients with the aim of increasing the vagal tone might improve depressive symptoms, but also exhibit beneficial effects on the cardio-vascular system”. Reference?
- “A pilot study found an im-provement of depressive symptoms, a reduced heart rate, and increased HRV in patients with MDD after HRV biofeedback but not in the control group”. Subjects, age, sex and number. You do not explain if they are women of a similar age to those in your intervention, or if they are older men, older women... etc.
- “Since the antidepressant effects of most therapies occur after weeks, the modulation of the ANS in MDD by respective challenges might serve as a biomarker to predict treatment response and also to allocate patients to specific therapies like HRV biofeedback”. Reference?
- study limitations well explained.
- It is unclear what the conclusion of the study is, since, being within the discussion and not separated into a specific section, it is not clear.
- The conclusion paragraph begins by talking about other studies carried out. I should after explaining those studies write a new paragraph just for the conclusion.
- The limitations of the study should be in a different section than the discussion.
REFERENCES
- A bibliographic manager should be used so that the citations are well inserted, homogeneous and in accordance with the standards recommended by the journal. It is important that the year appears in bold and the journal in italics. Check the references because many of them are wrong (names, rules of the references, etc.).
- Check some references.
- References will appear in alphabetical order according to APA 7 edition regulations.
Comments on the Quality of English Languagenon
Reviewer 2 Report
Comments and Suggestions for Authors
The present article evaluates the autonomic nervous system (ANS) response in patients with major depressive disorder (MDD) without coronary heart disease (CHD). The study is well conducted and provides important information on the role of ANS in MDD.
Some suggestions:
1. Introduction: The text does not explicitly delineate the rationale behind the study's focus on patients without CHD. While it is possible to deduce the reasons based on the presented information, it would be beneficial to explicitly state them in the text.
2. Lines 95-96: delete the sentence: “The initial cohort comprised 100 adult subjects (50 patients with 96 MDD and 50 healthy controls)”. Since exclusion criteria were mentioned earlier, repeating details about beta-blockers and atrial fibrillation is unnecessary.
3. Measurement of heart rate variability paragraph: It does not specify how "calm" is defined. Ideally, the environment should be standardized (e.g., dimly lit, quiet room with controlled temperature) to minimize external factors influencing pre-measurement relaxation.
4. In the absence of any consideration of gender disparities, remove the percentage of women from Table 1.
5. Discussion: Although the text mentions an attenuated ANS response in MDD, a more explicit explanation of how this relates to the response patterns observed during challenges (standing, SPB) would be helpful. Furthermore, I suggest highlighting the potential of using ANS modulation as a biomarker of treatment response in MDD.
6. It is suggested that a conclusion be included, in which the main findings are briefly reiterated, their importance emphasised, and future research directions proposed.
7. Follow the instruction for authors. For example, References must be numbered in order of appearance in the text.
Reviewer 3 Report
Comments and Suggestions for Authors
Neurodegenerative diseases constitute a major problem of public health that is associated with increased risk of mortality and poor quality of life. Malnutrition and poor habitual life are considered as a major problem that worsens the prognosis of patients suffering from neurodegenerative diseases, especially in patients with depression. In this aspect, the present article aimed to collect and summarize the available traditional clinical data as far as concern the clinical impact of MDD assessment in neurodegenerative diseases, highlighting on the heart disease symptom of ANS dysfunction. Although, the present study reported the possible roles that involved in the cardiac function. However, I do have some questions raised.
1. Authors should provide more information of HPA axis hyperactivity in the text. Lack of biochemical data is not easy to understand the ANS and cardiac dysfunction of subjects.
2. Authors should describe or target the potential HPA axis (hormone or reliable biomarker level) in Introduction and Discussion sections. And also focus on the interactions of ANS over activity and neuro-parameters to explain the brain-heart interaction in Results and Discussion sections.
3. Overall, this was an old but interesting topic, however lack of direct evidence to demonstrate the authors’ aim. Lack of the solid and direct data to convince reader that potential actions that affects the status of cardiac functions. The expression of Tables should be more precise. For a perspective aspect for disease prevention, some update potential biomarkers should be discussed. Thus, I do not recommend it in current status.
